# Incorporation of an Allogenic Cortical Bone Graft Following Arthrodesis of the First Metatarsophalangeal Joint in a Patient with Hallux Rigidus

**DOI:** 10.3390/life11060473

**Published:** 2021-05-24

**Authors:** Iva Brcic, Klaus Pastl, Harald Plank, Jasminka Igrec, Jakob E. Schanda, Eva Pastl, Mathias Werner

**Affiliations:** 1Diagnostic and Research Institute of Pathology, Medical University of Graz, 8010 Graz, Austria; 2Department for Orthopedic Surgery, Diakonissen Hospital Linz, 4020 Linz, Austria; klaus@pastl.at (K.P.); eva@pastl.at (E.P.); 3Graz Centre of Electron Microscopy, 8010 Graz, Austria; harald.plank@felmi-zfe.at; 4Division of General Radiology, Department of Radiology, Medical University of Graz, 8036 Graz, Austria; jasminka.igrec@medunigraz.at; 5Department for Trauma Surgery, AUVA Trauma Center Vienna-Meidling, 1120 Vienna, Austria; jakob.schanda@auva.at; 6Ludwig Boltzmann Institute for Experimental and Clinical Traumatology, 1200 Vienna, Austria; 7Austrian Cluster for Tissue Regeneration, 1200 Vienna, Austria; 8FB Pathology, Osteopathology, VIVANTES Netzwerk für Gesundheit, 13407 Berlin, Germany; mathias.werner@vivantes.de

**Keywords:** hallux rigidus, allogenic, cortical bone graft, screw, arthrodesis

## Abstract

Hallux rigidus is degenerative arthritis of the first metatarsophalangeal joint characterized by pain and stiffness in the joint with limitation of motion and functional impairment. Recently, bone grafts have been introduced in orthopedic procedures, namely osteosynthesis and arthrodesis. Allografts can induce bone formation, provide support for vascular and bone ingrowth and have a low risk of immunological rejection. A 52-year-old female patient with hallux rigidus underwent arthrodesis of the first metatarsophalangeal joint using Shark Screw^®^ made of allogenic cortical bone. Corrective surgery was performed after 10 weeks, and a 5 × 3 mm large part of the Shark Screw^®^ with the surrounding patient’s bone was removed. A histological evaluation revealed a vascularized graft with the newly formed compact lamellar bone fitting exactly to the cortical graft. The bone surface was lined by plump osteoblasts with osteoid production, and osteocytes were present in the lacunae. The arthrodesis of the first metatarsophalangeal joint using an allogenic cortical bone graft results in fast, primary bone healing without immunological rejection. This case suggests that the cortical allograft is a good and safe treatment option with an excellent graft incorporation into the host bone. However, as the literature evaluating the histology of different bone grafts is scarce, further high-level evidence studies with adequate sample sizes are needed to confirm our findings.

## 1. Introduction

Hallux rigidus is the most common degenerative arthritis of the first metatarsophalangeal joint (MTPJ). It is characterized by pain and stiffness in the joint with range-of-motion (ROM) limitation and functional impairment [1]. Nonsurgical management such as shoe modifications, use of oral nonsteroidal anti-inflammatory drugs (NSAIDs), corticosteroid injections, and physical therapy is used in the early stages of the disease to prevent or postpone surgical treatment. Patients with severe pain and functional impairment that are refractory to conservative treatment usually undergo surgery.

Several surgical procedures were used in hallux rigidus treatment [2,3]. The surgical approach is based on the degree of joint involvement, the extent of deformity visible on radiographs, the ROM limitations, and the patient’s activity level.

Bone healing is a complex physiological process aiming at the renewal of the biological and mechanical properties of the pre-existing bone [4,5]. It involves the interaction of different cell types, inflammatory cytokines, prostaglandins, growth factors, hormones, and vitamins. The secondary bone healing usually happens during the process of fracture with gap and callus formation. Primary bone healing, also known as direct healing, is rare and includes bone remodeling that occurs under stable conditions such as a fracture with rigid fixation (open reduction and internal fixation) without gap formation. The essential drivers for optimal bone healing in this process are angiogenesis and Haversian canal remodeling [6].

Cortical and trabecular bone allografts are used in different osseous reconstructive procedures such as fractures, tumor surgery, and osseous discontinuity [7,8]. Cortical allografts heal by osteoclastic resorption and the creation of new vessels followed by osteoblastic activity and the production of new bone [9]. The incorporation of a cancellous bone allograft is a process called “creeping substitution”, in which the grafted bone is simultaneously resorbed and replaced by newly formed bone from the host bone [10].

Herein, we present a female patient with hallux rigidus who underwent arthrodesis of the first MTPJ using Shark Screw^®^ made of allogenic cortical bone, with a histological and ultrastructural evaluation of the incorporation process after 10 weeks.

## 2. Materials and Methods

An informed consent form was obtained from the patient. The pain was assessed using the visual analogue scale (VAS). The function was assessed using the American Orthopedic Foot and Ankle Society Hallux Metatarsophalangeal Interphalangeal (AOFAS-HMI) scale.

### 2.1. Shark Screw^®^ Specific Data

The Shark Screw^®^ transplant (Surgebright GmbH, Austria) is a human cortical bone allograft for osteosynthesis and an alternative to metal or bioabsorbable devices in orthopedics and trauma surgery for adult and children [11]. It has a fine thread with a high rotational stability and accuracy of fit, and it offers the ideal conditions for the migration, proliferation, and differentiation of all bone cells. This allograft is remodeled without scars by the physiological bone metabolism, and, on average, within a year it is converted into the patient’s own bone.

### 2.2. Surgical Techniques

First surgery: The arthrodesis of the first MTPJ was performed using the cup and cone technique. After a dorsomedian skin incision, the joint capsule was opened lengthways. After placing the K-wires in the base of the proximal phalanx and in the first metatarsal, the articular surfaces were removed from the cartilage with the cup and cone. The joint surfaces were adjusted to one another in slight dorsiflexion and in a valgus position between 0 and 10 degrees and fixed with two crossing K-wires. The K-wires were used for drilling, thread cutting, and transplanting the Shark Screw^®^. The protruding part of the Shark Screw^®^ was set down at the bone level. 

Second surgery: A Chevron-like operation was performed. After a dorsomedian skin incision, K-wire was placed subcapital with a cutting guide. A reversed L osteotomy of the first metatarsal stem was performed. The distal first metatarsal section was valgized by pivoting it laterally. The exact position was temporarily held by the K-wires. After overdrilling and thread cutting, a Shark Screw^®^ was transplanted for osteosynthesis. The protruding part of the Shark Screw^®^ was set down at the bone level.

### 2.3. Light Microscopy

Toluidine blue stain was performed on 4-µm-thick sections after formalin fixation, undecalcified preparation, and methylmethacrylate embedding [12].

### 2.4. Scanning Electron Microscopy

Electron microscopy was carried out with a NOVA 200 dual beam microscope at 5 keV and 400 nA with the shortest dwell times possible to minimize beam damage. The samples were coated with a 10 nm carbon layer prior to any characterization. All images are unprocessed raw data.

## 3. Case Presentation

A 52-year-old female patient presented with a painful hallux rigidus of the right foot. In 2017, she complained of sharp pain when bending the toe. The MTPJ was painless and she started the treatment with NSAIDs. During the next two years, the pain increased in intensity and surgical treatment was suggested. The initial VAS and AOFAS-HMI score was 8.1 and 38.2, respectively. On the preoperative radiographs, the joint space was greatly narrowed due to the abrasion of the cartilage, and the head of the first metatarsal showed subchondral cysts, a degenerative metatarsal-sesamoid complex, corresponding to a Hallux rigidus grade III according to the Regnauld classification system (Figure 1A,B) [13].

The patient underwent arthrodesis of the first MTPJ with the cup and cone technique. Osteosynthesis was performed using two 35-mm-long Shark Screws^®^ with a diameter of 4.0 and 4.5 mm. The postoperative radiography showed two vertically oriented, crossed Shark Screw^®^ grafts extending from the metatarsal head to the base of the proximal phalanx (Figure 1C,D). However, the patient was not satisfied with the varus position of the big toe, and after 10 weeks a second surgery, a valgus osteotomy of the first metatarsus after Chevron, was performed proximally as a reversed L osteotomy with a short dorsal and long plantar leg. The great toe was brought into a light valgus. The osteosynthesis of the Chevron osteotomy was carried out with another 4.5 mm Shark Screw^®^. The control radiographs after reoperation showed a shortening of the proximal part of the medial screw in the metatarsal head and a newly introduced third screw in the distal metadiaphysis of the first metatarsal bone (Figure 1E,F). During this procedure, a 5 × 3-mm-large part of the Shark Screw^®^ with the surrounding patient’s bone was removed (Figure 1F inset), which contained the protruding shaft of the first metatarsal bone. Histologically, on Toluidine blue staining, the allograft was composed of lamellar bone with lamellae going parallel to each other without vital osteocytes in the lacunae (Figure 2). 

Interestingly, active osteoblasts lining the lamellar bone and newly formed vessels were observed in most of the Haversian canals (Figure 3A–D). In addition, we found osteoclasts removing the allograft bone with osteoblasts rebuilding the host bone (Figure 3). Moreover, directly on the screw surface, the newly formed compact lamellar bone was present with lamellae oriented perpendicular to the lamellae of the allograft bone, fitting exactly to the cortical graft (Figure 2).

The bone surface was lined by plump osteoblasts, which formed a close-fitting network of cells directly in contact with one another, with osteoid production (Figure 4A,B). Additionally, fully developed osteocytes were present in the lacunae (Figure 4A,B). In the whole specimen, no inflammatory infiltrate was observed. Ultrastructural analysis using scanning electron microscopy (SEM) confirmed the histological findings. Osteoclasts were found in the resorption lacunae within the graft and at the graft-host interface, the newly formed bone was lined with osteoblasts, and the osteocytes were inhabiting the lacunae (Figure 4C,D).

Six months after the reoperation, control radiographs showed the unchanged position of all three screws (Figure 5A,B). On a 13-month follow-up, the patient was without any symptoms. The VAS and AOFAS-HMI scores improved to 1.9 and 77.9 points, respectively. Control radiographs showed almost a complete resorption of the screws accompanied by a bony proliferation in the joint space resulting in the complete fusion of the first MTPJ (Figure 5C,D).

## 4. Discussion

Bone is constantly adapting and has the ability to heal without scarring. This dynamic process of rebuilding and adapting depends on an adequate vascular supply and metabolic activity [5,14]. In bone remodeling, the bone is resorbed by osteoclasts and newly formed by osteoblasts at the same location. These bone properties are used in bone grafting, as the bone grafts provide a scaffold to support and direct bone formation (osteoconduction) and vascular ingrowth [15].

Different internal fixation devices made of various materials have been used in orthopedic surgery to stabilize fractures and bone fragments including metallic (like steel and titanium) and bioabsorbable devices, as well as bone grafts (auto-, allo- and xenografts) [2,16,17,18]. The advantage of the sterilized bone graft is the absence of a second surgery for the device removal, a compatibility of the graft with radiology (CT and MRI), and the absence of an adverse tissue reaction or inflammation. Furthermore, cortical bone grafts have better mechanical properties when compared to cancellous bone grafts and can therefore provide better and immediate structural support.

Autogenous bone grafts have strong osteoconductive properties, can induce osteoinduction, and may serve as a source of osteoprogenitor cells (osteogenesis). Therefore, they are regarded as the gold standard in the treatment of various conditions like fracture and nonunion [17,18,19,20]. In addition, the risk of infection and immunological rejection is low. The major disadvantages include their limited availability, large hematomas, and possible morbidity at the donor site. Similarly, allografts have good osteoconductive characteristics as well as a low risk of rejection. However, osteogenicity and osteoinductivity are considered to be low as they lack viable cells [17,18,21,22]. Nevertheless, their advantages include accessibility in different sizes and shapes and no donor site morbidity.

The factors that are crucial for allograft incorporation in the host bone are revascularization with a new bone formation and stable osteosynthesis with a good bone-graft union. In a study using cortical allografts from sheep femora, a new bone formation was seen at the superficial area of the intercortical region or the graft interface with revascularization [10]. In our patient with osteosynthesis, a primary healing process of the cortical bone allograft was observed: the graft was vascularized by the ingrowth of vessels into the majority of Haversian canals, and bone remodeling was in progress including osteoclastic and osteoblastic activity. This resulted in the creation of new host bone at the bone-graft interface without signs of immunological rejection after 10 weeks. 

To the best of our knowledge, this is the first reported case of hallux rigidus treated with an arthrodesis of the first MTPJ using an allogenic cortical bone graft. However, as the literature evaluating different bone grafts is scarce, further high-level evidence studies are needed to confirm our findings.

## 5. Conclusions

We describe the incorporation of an allogenic cortical bone graft after arthrodesis of the first MTPJ in a patient with hallux rigidus. The result was the alleviation of symptoms and a fast, primary bone healing without signs of rejection. This case suggests that cortical allografts are a good and safe treatment option with an excellent graft incorporation into the host bone.

## Figures and Tables

**Figure 1 life-11-00473-f001:**
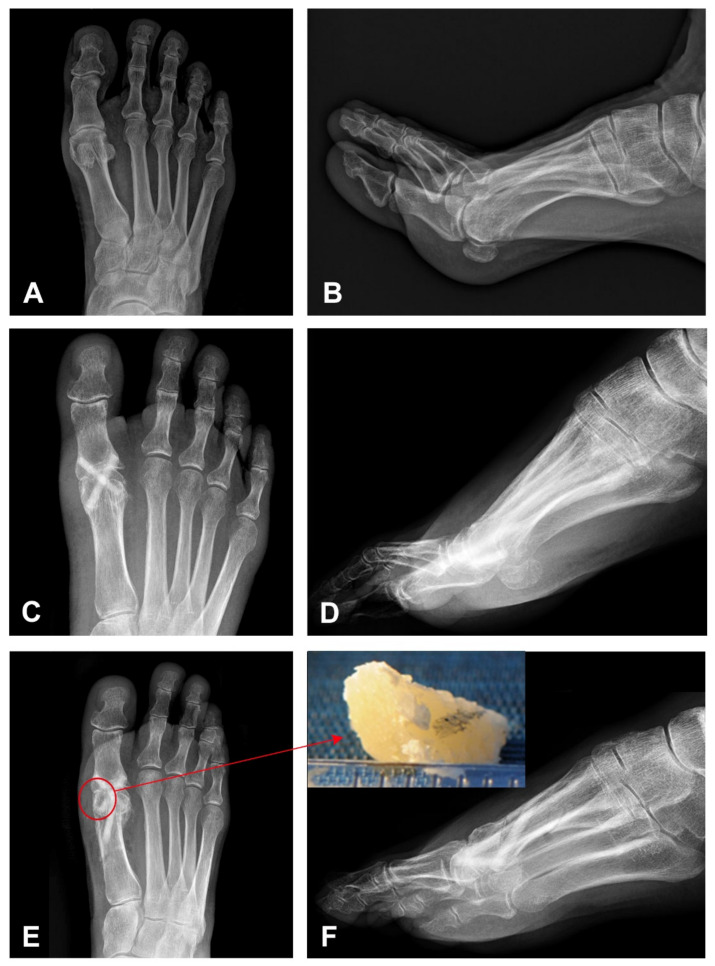
Initial and follow-up imaging findings on anterior-posterior (AP) and latero-lateral (LL) projections. (**A**,**B**) Radiograph of the right foot shows osteoarthritis of the 1. metatarsophalangeal joint of the great toe with small bone spurs and a joint space narrowing corresponding to Hallux rigidus. (**C**,**D**) Control radiography directly following arthrodesis of the 1st metatarsophalangeal joint of the great toe with two vertically oriented screws extending from the metatarsal head to the base of the proximal phalanx. (**E**,**F**) Postoperative radiography 10 weeks following arthrodesis of the 1st metatarsophalangeal joint of the great toe after the second correcting operation (a valgus osteotomy of the first metatarsus after Chevron). Initial bridging of the joint space medially is demonstrated. In addition, an adaptation of the position of the metatarsus with a shortening of the proximal part of the medial screw in the metatarsal head (red circle) and a newly introduced third screw in the distal metadiaphysis of the 1. metatarsal. Inset shows 5 × 3 × 3-mm-large part of the Shark Screw^®^ with the surrounding patient’s bone removed during the reoperation, containing the protruding shaft of the metatarsal bone.

**Figure 2 life-11-00473-f002:**
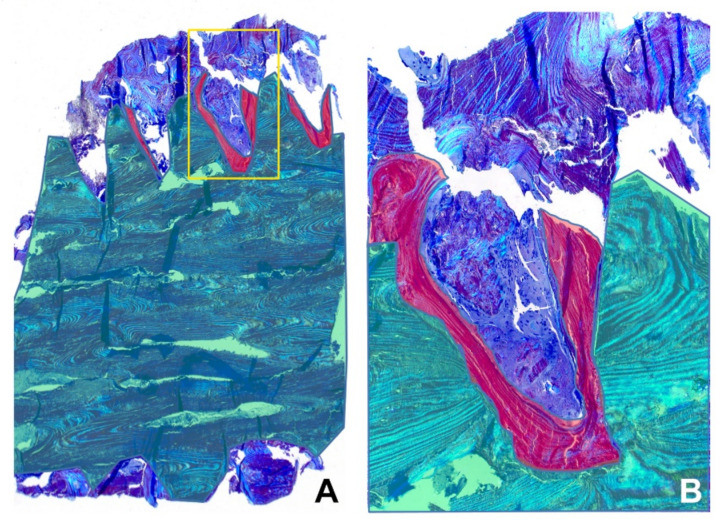
Histological findings highlighted in green and red color. (**A**) Low power of the whole specimen. The yellow rectangle shows the host-graft interface. (**B**) A higher view of the host-cortical graft interface marked with a yellow rectangle in A. (Toluidine blue stain). Color legend - Green: cortical graft composed of lamellar bone, the lamellae run from left to right. Red: newly formed bone fitting exactly to the cortical graft surface, the lamellae run tangential to the graft. Blue: host bone.

**Figure 3 life-11-00473-f003:**
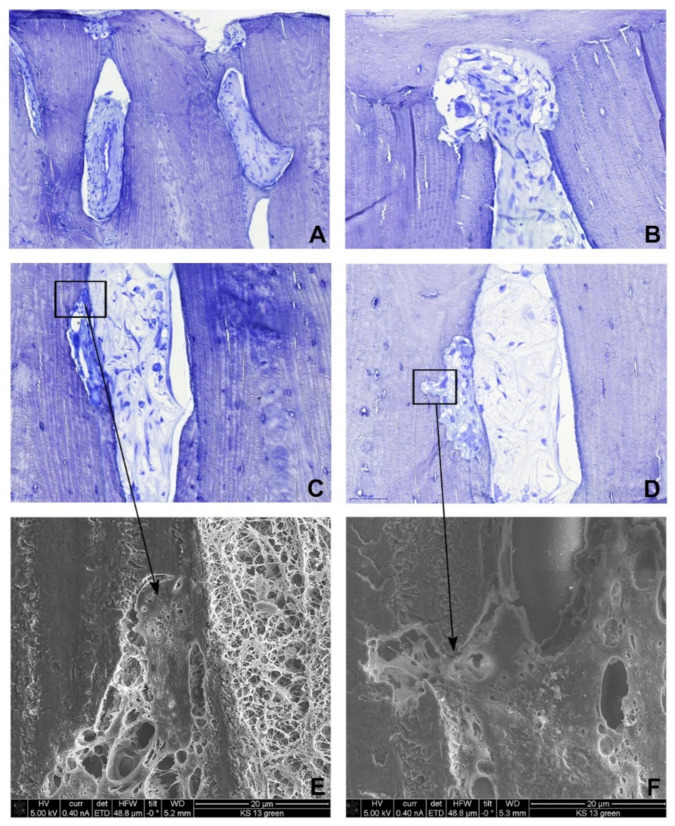
Histological (toluidine blue stain) and ultrastructural findings. (**A**) Newly formed vessels within the Haversian canals of the screw. (**B**–**D**) Osteoclastic activity (**B**) at the host-graft interface and (**C**,**D**) within the screw. (**E**,**F**) Scanning electron microscopy shows osteoclasts shown in (**C**,**D**) (rectangles) in the resorption lacunae of the graft.

**Figure 4 life-11-00473-f004:**
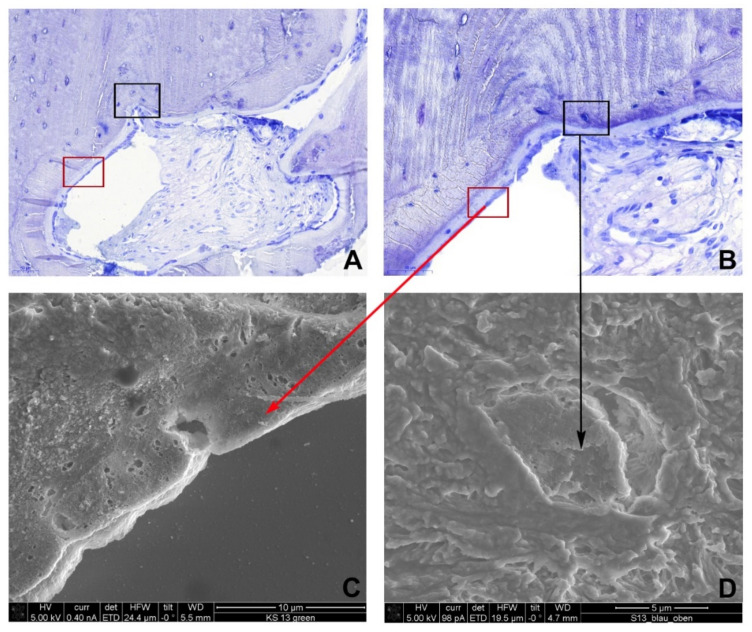
Histological (toluidine blue stain) and ultrastructural findings. (**A**,**B**) The host-graft interface shows a new bone formation fitting exactly to the cortical graft. The bone surface is lined with plump osteoblasts (red rectangle), and osteocytes (black rectangle) are present in the lacunae. Scanning electron microscopy shows (**C**) osteoblasts at the bone surface and (**D**) one osteocyte in the lacunae.

**Figure 5 life-11-00473-f005:**
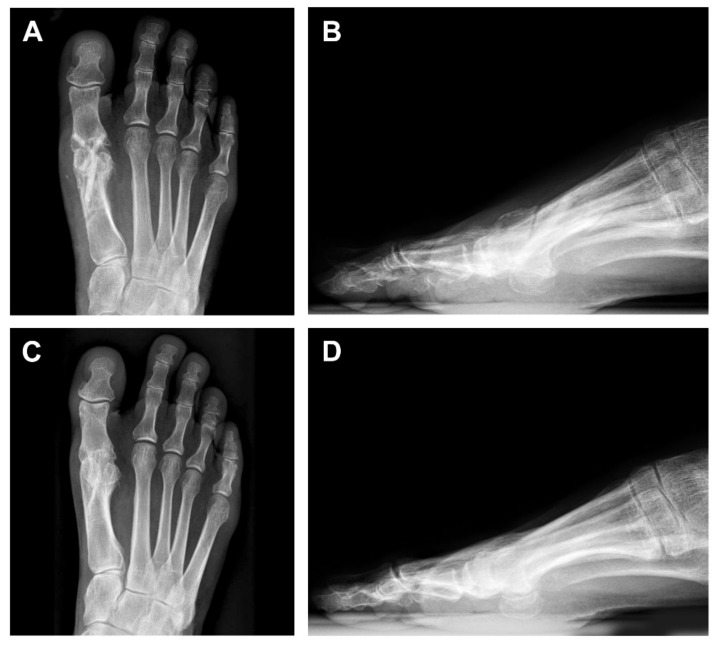
Imaging findings on anterior-posterior (AP) and latero-lateral (LL) projections after reoperation. (**A**,**B**) The control radiography six weeks after reoperation shows the unchanged position of all three screws. (**C**,**D**) On follow-up, 13 months after the operation, the control radiograph shows a complete remodeling of the screws accompanied by a bony proliferation in the joint space resulting in the complete fusion of the 1st metatarsophalangeal joint.

## Data Availability

Not applicable.

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
