# Peer review of "Incorporation of an Allogenic Cortical Bone Graft Following Arthrodesis of the First Metatarsophalangeal Joint in a Patient with Hallux Rigidus"

_life, 2021, doi:10.3390/life11060473_

Round 1

Reviewer 1 Report

Dear Authors, I think that the case report that you present is scientifically sound, well written and presented. I commend you for the work 

Author Response

Dear Authors, I think that the case report that you present is scientifically sound, well written and presented. I commend you for the work

Response: Thank you.

Reviewer 2 Report

Overall, it is a good and well-written article.

Abstract is correct, but I recommend not to use abbreviations and to divide it into sections.

The introduction is correct, but in lines 41-42 you have a confusing sentence "the extent of deformity on radiographs visible on radiographs", please modify or clarify.

In the methodology, I suggest including Shark Screwâspecific data and ethical data pertaining to the research, e.g. the patient's signature of informed consent for the data to be published.

Regarding the description of the clinical case, it is well documented and with images that help to understand the case. I suggest that in lines 94 and 99-101 the surgical techniques performed should be briefly defined, for example in brackets, or if you prefer, these definitions could be added in the methodology section.

The discussion and conclusion are correct, only the first and last sentence of the conclusion (“To the best of our knowledge, this is the first case of a hallux rigidus treated with an arthrodesis of the first MTPJ using an allogenic cortical bone graft” “However, as the literature evaluating different bone grafts is scarce, further high-level evidence studies are needed to confirm our findings.”) I think it is more correct to add them at the end of the discussion.

Author Response

Overall, it is a good and well-written article.

Abstract is correct, but I recommend not to use abbreviations and to divide it into sections.

Response: We changed the abbreviations. We structured the abstract as recommended by the journal: “ The abstract should be a single paragraph and should follow the style of structured abstracts, but without headings”.

The introduction is correct, but in lines 41-42 you have a confusing sentence "the extent of deformity on radiographs visible on radiographs", please modify or clarify.

Response: The sentence has been corrected.

In the methodology, I suggest including Shark Screw specific data and ethical data pertaining to the research, e.g. the patient's signature of informed consent for the data to be published.

Response: The Shark Screw data (with citation) and the “informed consent form” statement are included in the Methodology.

Regarding the description of the clinical case, it is well documented and with images that help to understand the case. I suggest that in lines 94 and 99-101 the surgical techniques performed should be briefly defined, for example in brackets, or if you prefer, these definitions could be added in the methodology section.

Response: Both surgical techniques are described in the Methodology section.

The discussion and conclusion are correct, only the first and last sentence of the conclusion (“To the best of our knowledge, this is the first case of a hallux rigidus treated with an arthrodesis of the first MTPJ using an allogenic cortical bone graft” “However, as the literature evaluating different bone grafts is scarce, further high-level evidence studies are needed to confirm our findings.”) I think it is more correct to add them at the end of the discussion.

Response: As suggested, we added  these two sentences at the end of the Discussion, and adapted the conclusions.

Reviewer 3 Report

The Author’s present a case report of a patient with hallux rigidus who  underwent fusion of the first MTPJ using allogenic cortical bone screws fixation with the histological and ultrastructural evaluation of the incorporation process after 10 weeks for a reoperation.

The topic is original, interesting, and the manuscript is well written. Despite this, some modifications are needed.

Line 76: please, cite the classification used to establish the grade of hallux rigidus and explain why it is a grade III hallux rigidus

Figure 1: a complete set of radiographs is necessary to allow the reader to evaluate the images. The lateral view radiographs are missing. Please, add lateral views of the images.

Line 98: the postoperative radiographs apparently show a varus ipercorrection of the hallux so maybe the Author’s should change the sentence with “the patient was not satisfied with the varus position of the big toe”

Methods and Results.  A preoperative and postoperative hallux functional score is missing in the manuscript. It should be added to scientifically evaluate the results of the procedure presented.

Author Response

The Author’s present a case report of a patient with hallux rigidus who  underwent fusion of the first MTPJ using allogenic cortical bone screws fixation with the histological and ultrastructural evaluation of the incorporation process after 10 weeks for a reoperation.

The topic is original, interesting, and the manuscript is well written. Despite this, some modifications are needed.

Line 76: please, cite the classification used to establish the grade of hallux rigidus and explain why it is a grade III hallux rigidus

Response: We included the classification, the description and the citation.

Figure 1: a complete set of radiographs is necessary to allow the reader to evaluate the images. The lateral view radiographs are missing. Please, add lateral views of the images.

Response: We included the lateral view radiographs (now included in Figure 1 and 5) with the description. In addition, we updated the Number of Figures in the main manuscript.

Line 98: the postoperative radiographs apparently show a varus hipercorrection of the hallux so maybe the Author’s should change the sentence with “the patient was not satisfied with the varus position of the big toe”

Response: This was corrected in the manuscript.

Methods and Results.  A preoperative and postoperative hallux functional score is missing in the manuscript. It should be added to scientifically evaluate the results of the procedure presented.

Response: We added the preoperative and postoperative functional scores in the manuscript.